# Real-Life Effectiveness of Mepolizumab on Forced Expiratory Flow between 25% and 75% of Forced Vital Capacity in Patients with Severe Eosinophilic Asthma

**DOI:** 10.3390/biomedicines9111550

**Published:** 2021-10-27

**Authors:** Angelantonio Maglio, Carolina Vitale, Simona Pellegrino, Cecilia Calabrese, Maria D’Amato, Antonio Molino, Corrado Pelaia, Massimo Triggiani, Girolamo Pelaia, Cristiana Stellato, Alessandro Vatrella

**Affiliations:** 1Department of Medicine, Surgery and Dentistry “Scuola Medica Salernitana”, University of Salerno, 84100 Salerno, Italy; amaglio@unisa.it (A.M.); carolinavitale.med@gmail.com (C.V.); pellegrino.simona3@gmail.com (S.P.); mtriggiani@unisa.it (M.T.); cstellato@unisa.it (C.S.); 2Department of Experimental Medicine, University of Campania “Luigi Vanvitelli”, 80100 Naples, Italy; cecilia.calabrese@unicampania.it; 3Department of Clinical Medicine and Surgery, University of Naples “Federico II”, 80100 Naples, Italy; marielladam@hotmail.it (M.D.); molinotonio@libero.it (A.M.); 4Department of Experimental and Clinical Medicine, University of Catanzaro “Magna Graecia”, 88100 Catanzaro, Italy; pelaia.corrado@gmail.com (C.P.); pelaia@unicz.it (G.P.)

**Keywords:** severe asthma, mepolizumab, eosinophils, lung function, FEF25-75, ACT, airway inflammation, small airways, anti-IL5, oral corticosteroid

## Abstract

Severe eosinophilic asthma (SEA) is associated with high peripheral blood and airway eosinophilia, recurrent disease exacerbations and severe airflow limitation. Eosinophilic inflammation is also responsible for small airway disease (SAD) development. SEA patients experience poor disease control and response to standard therapy and are prime candidates for anti-IL5 biologicals, such as mepolizumab, but the effect of treatment on SAD is unclear. We investigated the effect of mepolizumab on lung function in SEA patients, focusing on SAD parameters, and searched for an association between patients’ phenotypic characteristics and changes in small airways function. In this real-life study, data from 105 patients with SEA were collected at baseline and after 6, 12 and 18 months of mepolizumab treatment. Along with expected improvements in clinical and lung function parameters brought by Mepolizumab treatment, FEF2525-75% values showed a highly significant, gradual and persistent increase (from 32.7 ± 18.2% at baseline to 48.6 ± 18.4% after 18 months) and correlated with ACT scores at 18 months (r = 0.566; *p* ≤ 0.0001). A patient subgroup analysis showed that changes in FEF25-75% values were higher in patients with a baseline peripheral blood eosinophil count ≥400 cells/μL and oral corticosteroid use. Mepolizumab significantly improves small airway function. This effect correlates with clinical benefits and may represent an accessible parameter through which to evaluate therapeutic response. This study provides novel insights into the phenotypic characteristics associated with the improved functional outcome provided by mepolizumab treatment.

## 1. Introduction

Asthma is a chronic respiratory disease characterized by airway inflammation, variable airflow limitation and bronchial hyperresponsiveness. This widespread airway disorder affects over 300 million people worldwide and its prevalence is increasing. Although the majority of asthmatic patients can achieve good control over their symptoms using standard treatments, a minority of patients experience poor disease control, characterized by low quality of life (QoL) and high rates of exacerbations, hospitalizations and mortality [1,2]. Asthma exacerbations are defined as acute or sub-acute episodes of progressive worsening of respiratory symptoms and lung function that require a change in treatment, and can be classified as moderate or severe, based upon the acute clinical presentation and treatment needed. Patients requiring GINA step 4 or 5 treatment (e.g., medium- or high-dose inhaled corticosteroids with a second controller; maintenance oral corticosteroids) to prevent their asthma from becoming uncontrolled, or those remaining uncontrolled despite treatment, are considered affected by severe asthma. Rather than a single pathologic entity, asthma is currently thought to be a complex disease, characterized by heterogeneous traits with regard to etiology, triggers, inflammatory patterns, clinical manifestations and therapeutic responses [3]. Among the several different asthma phenotypes, eosinophilic inflammation occurs in more than 50% of patients with either atopic or nonatopic asthma. Severe eosinophilic asthma (SEA) is a disease subtype characterized by high numbers of eosinophils in both peripheral blood and airways, associated with recurrent disease exacerbations and severe airflow limitation [4]. SEA is sustained by type-2 (T2) inflammatory responses orchestrated by different cells and mechanisms: in atopic asthma, T helper 2 (Th2) lymphocytes drive an eosinophilic response based on an antigen-dependent mechanism; in non-atopic, late-onset asthma, the release of alarmins, induced by air pollution and viruses, activates group 2 innate lymphoid cells (ILC2), which release large amounts of the pro-eosinophil cytokine interleukin-5 [5,6].

Eosinophils are very sensitive to the anti-inflammatory action of glucocorticoids (GC) through direct and indirect mechanisms, including the inhibition of the expression of eosinophilic cytokines and chemokines [7]. However, SEA is characteristically GC-resistant, indicating the likely existence of multiple and heterogeneous mechanisms overriding molecular pathways conveying critical GC anti-inflammatory response [8]. Targeted therapy against IL-5 has fulfilled a crucial, previously unmet need for the treatment of GC-dependent SEA patients who are greatly affected by the side-effects of oral corticosteroid (OCS) therapy [9]; to this end, the OCS-sparing effect offered by anti-IL-5 biologics greatly contributes to patients’ overall improvement.

Eosinophilic inflammation is also closely related to small airway disease in asthma. Small airway dysfunction is highly prevalent in the asthmatic population, across all degrees of severity, particularly in patients with severe disease [10]. Patients with small airway disorder may experience poor disease control with poor response to inhalation therapy [11] and may thus benefit from anti-IL5 treatment. Therefore, the assessment of small airways impairment should be an important step in the management of severe asthmatic patients as well as in the evaluation of response to biological therapy.

Mepolizumab, a fully humanized anti-mouse IgG1/k monoclonal antibody, targets the eosinophilic subtype of severe asthma. The pharmacological efficacy of mepolizumab is due to the neutralization of the biological activity of IL-5, which prevents the terminal differentiation of eosinophil progenitors and reduces the output of eosinophils from bone marrow, resulting in a normalization of blood eosinophilia [4]. In large randomized controlled trials (RCTs), the effect of mepolizumab on blood eosinophils count was associated with a significant reduction in the intake/dosage of oral corticosteroids (OCS), a reduction of exacerbations and an improvement in symptom control with a very good tolerability and safety profile [1,2,3]. Along with its clinical benefits, mepolizumab demonstrated a significant effect on pulmonary function, although the results are less consistent [12,13,14].

The main purpose of this study was to investigate the potential effect of mepolizumab on forced expiratory flow at 25%–75% (FEF25-75) of forced vital capacity (FVC), and whether this effect is related to other clinical benefits. With this aim, we explored the potential usefulness of FEF25-75, a measure of small airway dysfunction widely performed in everyday clinical practice, as an adjunctive tool to assess response to mepolizumab therapy. Furthermore, we also assessed the possible phenotypic characteristics associated with a better functional outcome.

## 2. Materials and Methods

### 2.1. Study Design

This was a retrospective study that analyzed 105 patients with severe eosinophilic asthma treated with mepolizumab and referred to four centers in southern Italy. All the patients received a diagnosis of severe asthma according to the European Respiratory Society (ERS)/American Thoracic Society (ATS) guidelines [15] and were assessed for eligibility for mepolizumab treatment according to the Italian Drug Agency (AIFA)’s prescription criteria. Mepolizumab was administered subcutaneously at a dosage of 100 mg every 4 weeks and patients who had completed at least 6 months of therapy were considered for the analysis. The clinical, functional and biological data of included patients were recorded on a common database. Follow-up assessment occurred at 6, 12 and 18 months of treatment with mepolizumab. All the patients gave informed consent for the use of their personal data. This observational study was undertaken in accordance with the Helsinki Declaration and was approved by the Local Ethical Committee.

### 2.2. Data Collection

The following data were recorded at baseline: demographic characteristics, smoking history, BMI, duration of asthma, age of asthma onset, atopic status, presence of rhinitis, sinusitis, nasal polyposis and other comorbidities (obesity, gastroesophageal reflux), blood eosinophil count, total IgE level, fractional exhaled nitric oxide (FeNO) and pulmonary function tests, Asthma Control Test (ACT) score, number of exacerbations and asthma medications.

### 2.3. Asthma-Specific Outcome Variables

The clinical, biological and functional outcomes were collected 6, 12 and 18 months after the first injection of mepolizumab.

Clinical Outcome: ACT score, number of exacerbations, need of maintenance systemic corticosteroid. Exacerbations taken into account were those with worsening of symptoms, requiring oral corticosteroids for at least three days.

Biological outcome: blood eosinophils count and FeNO.

Pulmonary function tests: FEV1%, FVC%, FEV1/FVC% and FEF25-75%.

### 2.4. Statistical Analysis

The statistical analysis was performed using Prism Version 8.2.1 (Graphpad Software Inc., San Diego, CA, USA). The data were reported as mean and standard deviation (SD) for normally distributed data and median and interquartile range (IQR) for non-normally distributed data. The categorical variables were considered as the number of cases and percentages. The comparisons of continuous variables were performed using the t-test or the Wilcoxon signed-rank test, as appropriate, in order to assess the difference between the patient’s status before and after treatment. The comparisons between different sub-groups of patients were performed using Mann–Whitney-U test. The categorical variables were compared using Fisher’s exact test. Spearman’s rank correlation coefficient was calculated to assess the association between variables and a linear regression analysis was developed. Probability values of <0.05 were considered to be statistically significant.

## 3. Results

### 3.1. Baseline Characteristics of Study Population

Our retrospective observational study included 105 adult patients with severe eosinophilic asthma, according to the ERS/ATS definition [15], treated with mepolizumab for at least six months. The data on some clinical and biological features were not available for all the patients at the time of collection due to the retrospective nature of the study (baseline demographic, clinical, functional and inflammatory characteristics of the recruited patients are reported in Table 1).

The patients were more frequently female (67 females-63.8%), with a mean age of 58.5 ± 11.0 years old and a mean BMI above the normal range of 27.4 ± 4.2 kg/cm^2^. According to the ERS/ATS definition of severe asthma, all the patients required high dosages of inhaled corticosteroids (ICS) and long-acting beta agonist (LABA) combinations. Furthermore, 60% of the patients received additional bronchodilator treatment consisting of long-acting muscarinic antagonist (LAMA), 51.43% required continuous intake of oral corticosteroids (OCS) and 3.8% used leukotriene receptor antagonists (LTRA). Despite maximal therapy, most patients (84%) had uncontrolled asthma defined by a baseline ACT score <20 (according to GINA defined criteria). Moreover, they had a history of asthma complicated by frequent exacerbations (mean SD 4.2 ± 2.5). All the patients had eosinophil counts in peripheral blood of at least 150 cells/μL at baseline or at least 300 cells/μL within the previous 12 months, with a median (IQR) eosinophil count at baseline of 500 (350–717). Among all the enrolled patients, 55% were defined as allergic based on skin prick test (SPT) positivity and the most reported comorbidity was chronic rhinosinusitis (49.5%) with or without nasal polyposis (33.3%). Regarding lung function, the mean (±SD) baseline FEV1 and FEF25-75 was 63.7 ± 17.9 and 32.7 ± 18.2 of predicted value respectively.

Moreover, we detected a positive correlation between blood eosinophils counts and FEV1% of predicted (r = 0.274; *p* = 0.005), while no correlation was found between blood eosinophils and ACT score (r = 0.110; *p* = 0.287) and FEF25-75% of predicted (r = 0.037; *p* = 0.715) at baseline (Figure 1).

### 3.2. Response to Mepolizumab

#### 3.2.1. Clinical Effects

The significant clinical benefit offered by add-on therapy with mepolizumab in the studied SEA cohort was demonstrated by an improvement in symptom control score, a decrease in exacerbation rates and the occurrence of OCS-sparing effects. As a readout of symptom control, the ACT score significantly increased from a below-control baseline value of 14.5 (11; 17.7) (Median (IQR)) to near-threshold value for control of 19 (17; 22.5) after 6 months (ACT score for asthma control is >20), crossing the control threshold at 18 months’ follow-up, reaching a value of 23 (20; 24) (Figure 2, Panel a). On these bases, the percentage of patients with controlled asthma increased significantly and progressively from 17% at baseline to 46.8% and 72.4% after 6 and 18 months of add-on therapy, respectively (Figure 2, Panel b). Along with improved control, considerable reduction of the asthma exacerbation rate was registered. The annualized ratio of exacerbation dropped from 4.2 ± 2.5 to 1.1 ± 1.6 after one year of mepolizumab treatment (Figure 2, Panel c), with 36 patients exacerbation-free at 12 months of treatment follow-up. Finally, a significant OCS sparing effect was also demonstrated in our study as the percentage of patients requiring OCS therapy decreased significantly, from 51.4% at baseline to 19% at 6 months’ follow-up, with a further reduction in the following months (11.4% at 18 months’ follow-up) (Figure 2, Panel d; Appendix A).

#### 3.2.2. Biological Effects

After 6 months of add-on mepolizumab therapy, blood eosinophil count dropped from 500 cells/μL (350–717) [median (IQR)] to 80 cells/μL (40–130). This expected effect was sustained over time, reaching 50 cells/μL (30–100) after 18 months of treatment (Figure 3, Panel a). In 73 out of 105 patients, FeNO was measured with baseline values of 41 (24–61.5) ppb. The FeNO levels were unchanged after 6 months of treatment but decreased significantly on the 12 and 18 month follow-up assessments to 37 (23–53) ppb and 33 (21.5–44) ppb, respectively (Figure 3, Panel b; Appendix A).

#### 3.2.3. Effects on Pulmonary Function Tests

Treatment with mepolizumab determined a positive impact on lung function. Significant effects were already measurable according to all spirometric parameters after 6 months of add-on therapy. In particular, the mean percentage value for FEV1, FVC and FEV1/FVC increased almost to their maximal improvement level after 6 months of therapy, without further significant improvement over the course of the observation period (Figure 4, Panel a–c). Interestingly, treatment with mepolizumab induced a more gradual and persistent increase in FEF25-75%. In fact, the mean value increased progressively during the study period from a baseline of 32.7 ± 18.2% to 40.8 ± 21.3% after 6 months, with statistically significant increases from the previous timepoint detected after 12 and 18 months of treatment (with values of 45.3 ± 21.1% and 48.6 ± 18.4%, respectively), indicating a slower but progressive improvement in small airway obstruction (Figure 4, Panel d).

We also evaluated correlations between clinical scores (ACT) and pulmonary function tests (Figure 5) with regard to: FEV1_%_ at baseline (r = 0.359; *p* = 0.0003) and at 18 months’ follow-up (r = 0.400; *p* = 0.002), FVC% at baseline (r = 0.231; *p* = 0.023) and at 18 months’ follow-up (r = 0.255; *p* = 0.062); FEV1/FVC% at baseline (r = −0.100; *p* = 0.333) and at 18 months’ follow-up (r = −0.359; *p* = 0.068), FEF25-75% at baseline (r = 0.347 *p* = 0.0007) and at 18 months’ follow-up (r = 0.566; *p* = <0.0001) (Appendix A). A Pearson’s correlations comparison was also performed, with the following results: FEV1, FVC and FEV1/FVC correlation with ACT on baseline and after 18 months were not statistically significant (FEV1 vs. ACT, z: −0.342; *p*: 0.366; FVC vs. ACT, z: 1.237; *p*: 0.108; FEV1/FVC vs. ACT, z: −1.124; *p*: 0.13), while the comparison between FEF25-75 correlation with ACT on baseline and after 18 months was statistically significant (z: −1.997; *p*: 0.023).

#### 3.2.4. Efficacy of Mepolizumab across Different Subgroups of Asthmatic Patients

We then evaluated the study outcomes in different subgroups of patients according to age of asthma onset, allergic phenotype, baseline eosinophilic count and need for maintenance oral corticosteroids (mOCS). No difference was found in terms of treatment-related improvement in clinical, biological and pulmonary function outcomes according to age of asthma onset (<40 or ≥40 years), allergic phenotype defined on the basis of SPT results (data not shown), and obesity.

To evaluate the response to mepolizumab according to baseline blood eosinophil level, we chose a cut-off of 400 cells/μL [16,17], based on the results obtained from the comparison between subgroups of patients with different eosinophils levels at baseline (between 150–300 cells/μL; between 300–399 cells/μL; ≥400 cells/μL; the graphs in Appendix A show no statistically significant difference among these subgroups after 12 months of treatment). Patients with blood eosinophils ≥400 cells/μL (63.8%) showed a higher degree of airway obstruction when compared to the subgroup with eosinophils <400 cells/μL, with a FEV1/FVC% significantly lower both at baseline and on follow-up assessments (Appendix A).

Moreover, patients with higher blood eosinophil levels reported a significantly greater improvement in ACT score compared to baseline both at 12 months (change in ACT of 8 (5–9) vs. 5 (1–8) *p* = 0.04) (Figure 6, Panel a) and 18 months’ follow-up (change in ACT of 8 (4–10) vs. 5 (2–7), *p* = 0.03). In this high-eosinophils subgroup, the improved symptom control after 12 months of treatment with mepolizumab was paralleled by a greater improvement both in FEV1 (12 (6.3–19) vs. 7 (0–12)% of predicted, *p* = 0.01) (Figure 6, Panel b) and FEF25–75 (13 (5–22.3) vs. 8 (3–12)% of predicted, *p* = 0.04) (Figure 6, Panel c). No difference was found between the two subgroups in the reduction of exacerbation rate. (Appendix A)

Along the same lines, = the subgroup of patients requiring mOCS demonstrated higher baseline blood eosinophil counts (560 (365–820) vs. 400 (330–600), *p* = 0.01) (Figure 7, Panel right) as well as higher levels of airway obstruction (FEV1/FVC ratio) at baseline and follow-up assessment when compared to the patients who did not need mOCS. In addition, the patients requiring mOCS showed a greater improvement in small airways obstruction compared to baseline both at 12 months (14 (4.8–23) vs. 7 (5–13)% of predicted, *p* = 0.04; Figure 7, Panel right) and 18 months (18 (9–34) vs. 11 (6–18.5)% of predicted, *p* = 0.04) of add-on therapy with mepolizumab (Appendix A).

## 4. Discussion

Our data provide novel insights into the established beneficial effects of mepolizumab on lung function, showing significant improvements in airflow in the small airways (FEF25-75), as well as in the large airways (FEV1), and the association of mepolizumab with specific subgroups of SEA patients with high eosinophil counts and OCS use.

Overall, our study aligns with findings reported by previous RCTs and recent real-life observations [18,19,20,21,22,23], demonstrating a significant improvement in lung function and positive effects on symptom control, exacerbation rate and OCS use in a cohort of patients with SEA. As expected, mepolizumab efficacy was associated with a very good safety and tolerability profile [4].

Although data regarding the clinical efficacy of mepolizumab in terms of symptoms, exacerbation rate and OCS sparing effect are strong, currently available findings regarding its therapeutic action on lung function are less evident. A recent Cochrane analysis [24] reported that anti-IL-5 monoclonal antibodies produced slight, but statistically significant, improvements in mean pre-bronchodilator FEV1, ranging from 0.08 L to 0.11 L [25,26]. With the use of mepolizumab, FEV1 increases reported by early RCTs were modest, and below the level of clinical relevance. Indeed, the results of the DREAM trial reported only a small FEV1 increment [27], whereas subsequent MENSA and MUSCA studies showed greater improvements [12,13]. Later on, other investigations carried out in real-life settings made it possible to document better results referring to lung function [18,19,20,21,22]. In our patient cohort, mepolizumab improved airflow limitation, as demonstrated by the increases in FEV1, FEV1/FVC ratio and FEF25-75. In particular, we detected higher increments of FEF25-75 values that were more gradual and of greater amplitude than those reported for FEV1. These data highlight a significant effect of mepolizumab on airflow limitation in the peripheral airways. Indeed, in real-world settings, FEF25-75 is considered to be the most commonly used spirometric indicator of small airway patency in real-life clinical practice [28]. Within the context of the Severe Asthma Research Program (SARP) project, it was observed that some severe asthmatic patients who had a more severe reduction of FEF25-75 also displayed more symptoms and a greater need for medication, as well as higher FeNO levels, total serum IgE concentrations, and bronchial hyperresponsiveness [29]. This evidence supports the hypothesis that in a subgroup of asthmatic patients, a low FEF25-75 value is an independent marker of asthma severity. Consistently with previously published data, FEF25-75 was the most impaired lung parameter in our study population at baseline, with 70% of the enrolled subjects being characterized by a result lower than 40% of the predicted FEF25-75 measures. FEF25-75 was also the parameter for which we recorded the most significant increase over time. In contrast with previous data obtained on asthmatic children [30], our analysis did not demonstrate a significant correlation between baseline values of peripheral eosinophils and FEF25-75% of predicted. Interesting data are provided by a recent study that established a negative correlation between FEF25-75% of predicted and sputum eosinophils count [31]. However, the association between blood eosinophilia and peripheral airway obstruction has not been reported before [32]. In basal conditions, the reduction of FEF25-75 is probably determined by various factors, one of which is eosinophilic inflammation. Eosinophilic inflammation is the potentially reversible component of airflow limitation, and for this reason anti-IL5 treatment could justify the improvement over time of FEF25-75 in the population with the highest peripheral eosinophil counts (Figure 6). The efficacy of mepolizumab against small airway obstruction was recently investigated in a prospective study through multiple breath nitrogen wash-out, a sensitive lung function test that is rarely available in common clinical practice [33]. More recently, a retrospective study, performed in a real-life setting, demonstrated a significant effect of mepolizumab on small airways by using FEF25-75 [34]. The positive impact of mepolizumab on small airway obstruction could be explained by its systemic delivery, allowing an adequate concentration of the drug at distal sectors of the respiratory tree, where eosinophilic inflammation is prominent [35,36]. These results provide new perspectives on the treatment of small airway disease in severe asthma, since systemic treatment would be able to overcome possible therapeutic failures related to the inhaled administration route. The improvement in lung function, and in small airway caliber in particular, may be a significant contributing factor to mepolizumab therapeutic response, and it could partly explain the clinical improvement of mepolizumab-treated patients (including a reduction in asthma symptoms and exacerbation rates). Although there are many factors supporting clinical improvement, the strong correlation we found between the ACT score and FEF25-75 at 18 months of follow-up is of particular interest. In fact, the percentage of patients overcoming the critical threshold of 20 in the ACT score, indicative of an adequate control of symptoms, increased from 17% at baseline to 72% after 18 months of add-on biological therapy with mepolizumab.

Validated treatment response criteria to mepolizumab remain are yet to be defined precisely. According to the recommendations of the British National Institute for Health and Care Excellence (NICE), an adequate therapeutic response can be provided by a reduction = in asthma exacerbation rates of at least 50%, or by a clinically significant decrease in continuous OCS use [37]. Although both these goals were met in our study, these outcomes alone may not be comprehensive enough to assess the efficacy of mepolizumab in a real-life context, given the wide heterogeneity of the population suffering from severe asthma. Indeed, not all patients are frequent exacerbators and, similarly, not all subjects require regular OCS therapy. Therefore, evaluation of the therapeutic response to mepolizumab cannot rely only on assessment of the above two outcomes, but should also take into account the improvements in both subjective conditions and lung function, which were investigated by our study.

We also planned to identify possible baseline features associated with better functional and clinical responses to mepolizumab treatment. These characteristics included age of asthma onset, atopic trait, blood eosinophil count and need for maintenance OCS. Interesting differences between OCS-dependent subjects and non-OCS dependent subjects were found. At baseline, the OCS-dependent subjects presented a greater blood eosinophil count and degree of peripheral airflow limitation than the non-OCS-dependent subjects. These results may suggest that the OCS-dependent patients, despite treatment with systemic steroids, had a more severe form of the disease. These data appear to be similar to the data obtained in previous research [38]. Both allergic status (defined on the basis of SPT positivity or negativity) and age of asthma onset (<40 or ≥40 years) were not shown to be associated with the reported improvements in clinical and functional outcomes, thereby corroborating the results of other real-life studies [21,39]. Similarly, we found no correlation between obese and non-obese patients. These findings seem to be in contrast with the data reported in previous research, which suggest that obesity could be a risk factor for severe asthma [40,41], with a reported prevalence of 60% of severe asthmatic patients [42]. The small number of obese patients in our study population may explain this lack of correlation. On the other hand, we found statistically significant differences in terms of better symptom control and spirometric parameters, regarding patients characterized by higher blood eosinophil counts and greater need for OCS therapy. Consistently with our present findings, it was previously demonstrated that the magnitude of response to mepolizumab is more closely related to blood eosinophilia than to sputum eosinophilia [14]. Several studies investigated the relationship between baseline blood eosinophil numbers and therapeutic response to mepolizumab using different cutoffs, thus obtaining similar results in terms of the reduction of exacerbation rates, but less conclusive findings for lung function and asthma control [27]. Although the universally recognized cutoff of blood eosinophilia for mepolizumab prescription is 150 cells/μL, coupled to the detection of at least 300 cells/μL during the previous 12 months, we evaluated the response to mepolizumab according to a higher eosinophil level, with a cutoff of 400 cells/microliter, and so we obtained two homogeneous populations to evaluate [43]. In order to identify specific prognostic features, stratifying patients between subgroups with more or less than 400 eosinophils is a strategy that has already been performed, and previous data show that the effect of mepolizumab generally appears to be greater in subgroups with greater eosinophil counts [44]. In particular, a threshold of 400 cells/μL may determine a subgroup of patients displaying a better functional response in terms of increases in both FEV1% predicted and FEF25-75% predicted. The greater beneficial effect on airway caliber, observed in patients with blood eosinophils ≥400 cells/μL, was also found to be associated with a greater increment of ACT score. These data suggest that the lung functional benefit provided by mepolizumab may represent a significant contributing factor to the clinical effectiveness of this biological drug.

Another interesting finding observed in our study was the progressively increasing improvement in FEF25-75, detected in the subgroup of patients requiring OCS maintenance therapy, when compared to subjects not needing regular OCS treatment. It is also noteworthy that the patient subgroup under continuous OCS therapy demonstrated higher baseline blood eosinophil counts, despite OCS treatment. For this cluster of patients, it can be proposed that the reduction of peripheral blood eosinophil numbers, observed at every time point of our study, could have been related to the anti-IL-5 activity exerted by mepolizumab within the context of glucocorticoid-resistant eosinophilic inflammation, and was thus not modifiable by OCS.

This study did feature some limitations. Given its retrospective and multicenter design, some datasets were smaller due to different clinical record-taking across participating centers. Furthermore, as a functional marker of peripheral airflow limitation, we used FEF25-75% predicted, whose role in identifying small airway dysfunction has been thoroughly debated in terms of reproducibility and sensitivity and can be operator-dependent, since flows at percentages of FVC strictly depend on the correct spirometric maneuver [45].

Our data indicate that despite these limitations, FEF25-75% predicted can be a valuable asset as a first-level assessment for SAD, since it is routinely performed in common clinical practice and could be used as a treatment response parameter. Moreover, these data appear to be even more significant, since they do not come from an RCT. Improvements in FEF25-75% predicted by Mepolizumab in a real-life context highlight the relevance of distal airways as targets for asthma control, since data in recent research showed that SAD affects 50–60% of individuals across the range of persistent asthma severity [10].

## 5. Conclusions

Even though mepolizumab has already proven to be very effective and well tolerated, results from real-life experiences show that this drug can induce better therapeutic effects than those reported by RCTs. Consistently with such findings, in this study we demonstrated that mepolizumab elicited long-lasting improvements in lung function, especially with regard to relevant increases in small airway caliber. This effect, closely related to the mepolizumab-mediated attenuation of eosinophilic inflammation, probably contributes in a decisive manner to persistent clinical benefits. It is difficult to reliably predict which patients will benefit the most from mepolizumab based on their baseline features. However, we identified subgroups of patients with SEA for whom the lung functional benefits appeared to be more significant, thus providing potentially important insights into our understanding of the phenotypic features associated with a better response in terms of clinical and functional outcomes.

## Figures and Tables

**Figure 1 biomedicines-09-01550-f001:**
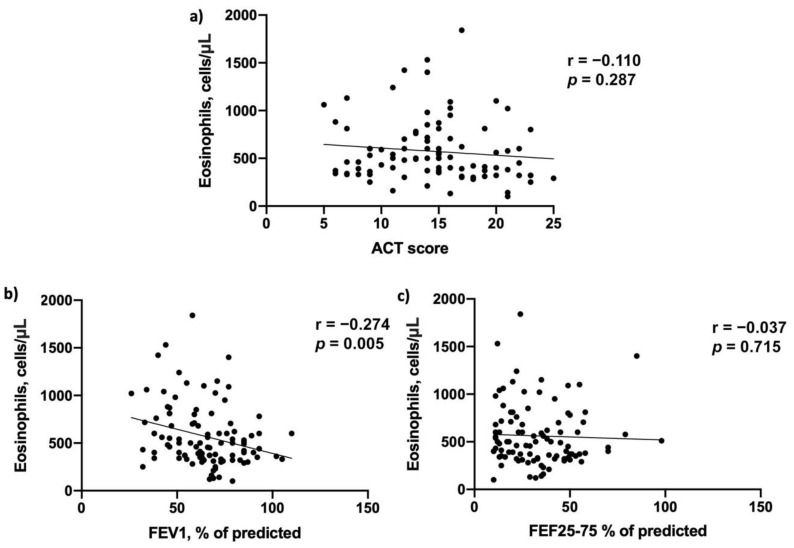
Baseline correlations between blood eosinophils count and (**a**) Asthma Control Test (ACT) score, (**b**) FEV1% of predicted and (**c**) FEF25-75% of predicted.

**Figure 2 biomedicines-09-01550-f002:**
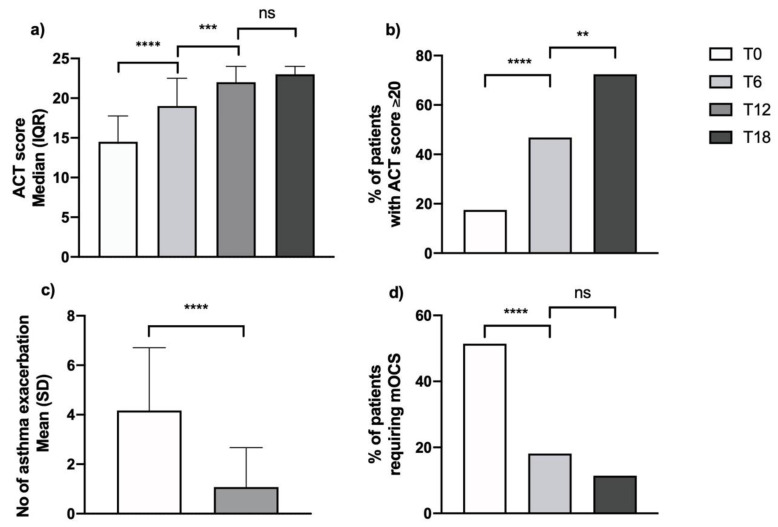
Effects on clinical parameters during treatment with mepolizumab on (**a**) ACT score; (**b**) % of patients with good asthma control; (**c**) asthma exacerbations; (**d**) % of patients requiring maintenance OCS (**, *p* = 0.002; ***, *p* = 0.0002; ****, *p* < 0.0001; ns, non-significant).

**Figure 3 biomedicines-09-01550-f003:**
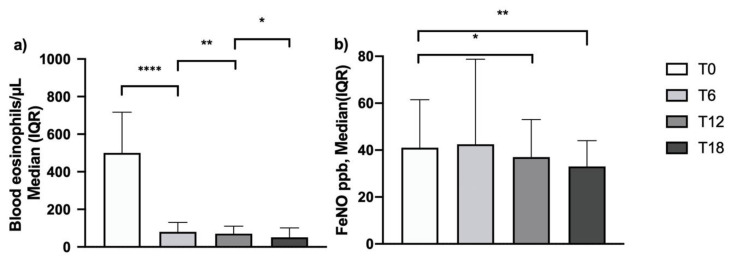
Effects of mepolizumab on blood eosinophilia (**a**) and FeNO (**b**) (*: *p* = 0.04; **: *p* = 0.004; ****: *p* < 0.0001) and FeNO (*: *p* = 0.01; **: *p* < 0.001; ns, non-significant).

**Figure 4 biomedicines-09-01550-f004:**
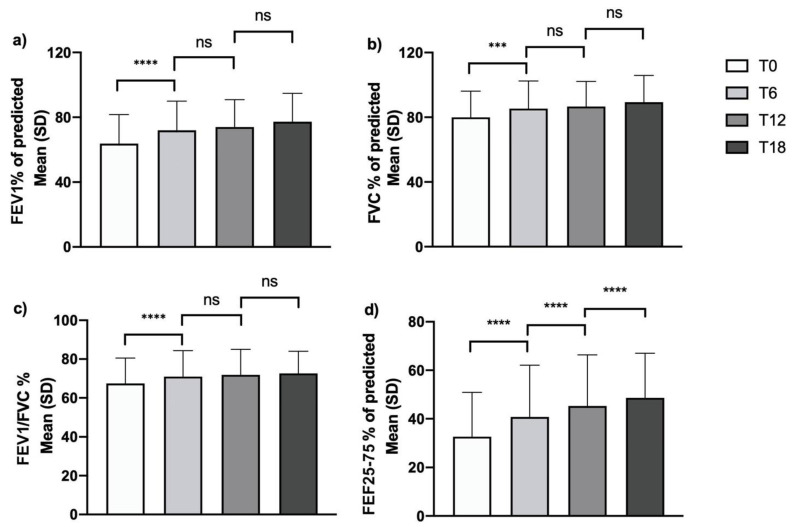
Effects of mepolizumab on pulmonary function tests: changes over time in (**a**) FEV1% of predicted; (**b**) FVC% of predicted; (**c**) FEV1/FVC%; (**d**) FEF25-75% of predicted (***: *p* = 0.001; ****: *p* < 0.0001; ns, non-significant).

**Figure 5 biomedicines-09-01550-f005:**
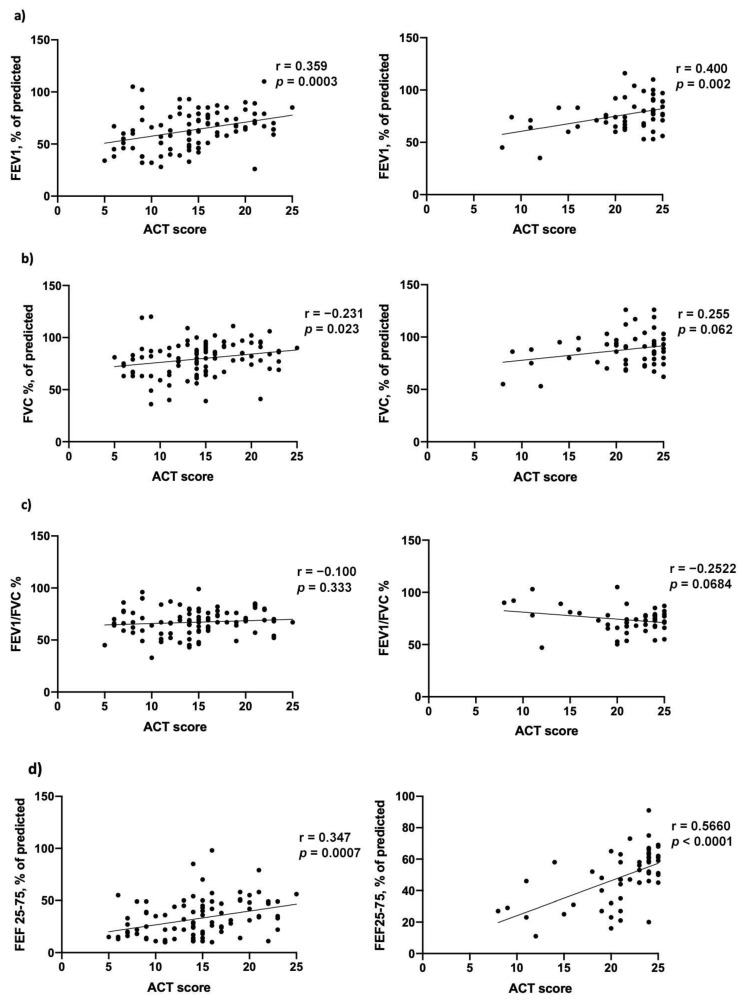
Correlation between Asthma Control Test (ACT) score and pulmonary function tests at baseline (**left**) and after 18 months of mepolizumab treatment (**right**), with regard to (**a**) FEV1% of predicted, (**b**) FVC% of predicted, (**c**) FEV1/FVC% and (**d**) FEF25-75% of predicted.

**Figure 6 biomedicines-09-01550-f006:**
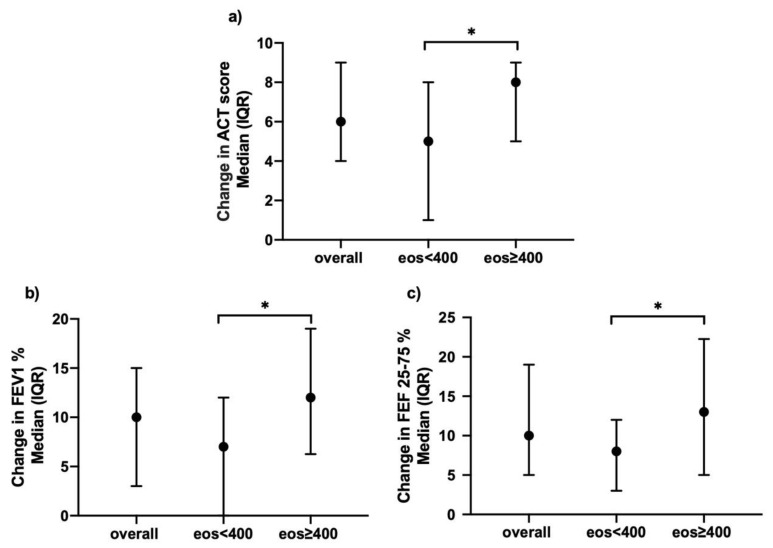
Effects of mepolizumab on (**a**) ACT; (**b**) FEV1%; (**c**) FEF25-75%, clustered upon level of blood eosinophilia, after 12 months of treatment (*: *p* < 0.05).

**Figure 7 biomedicines-09-01550-f007:**
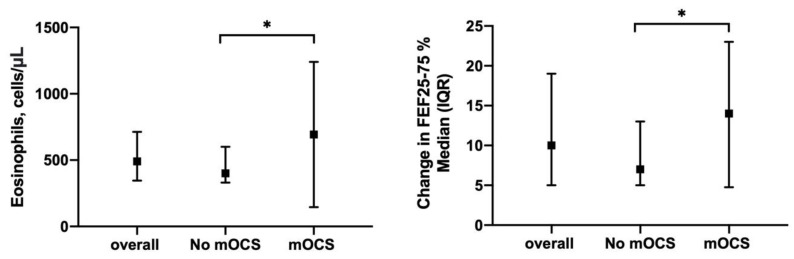
Sub-analysis of patients clustered upon need for maintenance oral corticosteroids (mOCS): (**Left**) shows baseline blood eosinophil counts between No-mOCS and OCS subgroups; (**Right**) shows change in FEF25-75% median after 12 months of treatment in the same subgroups (*: *p* < 0.05).

**Table 1 biomedicines-09-01550-t001:** Baseline characteristics of patients with severe eosinophilic asthma prior to mepolizumab treatment.

**Demographics (*n* = 105)**	
Age	58.5 ± 11.0
Gender (female)	67 (63.8%)
Smoking habit (*n* = 102)	
Never	58 (56.86%)
Smokers	19 (18.63%)
Ex-smokers	25 (24.51%)
Body Mass Index kg/m^2^, *n* = 105	27.4 ± 4.2
Asthma duration, years, *n* = 93	21.5 ± 13.7
Age of asthma onset, years, *n* = 93	37.4 ± 14.3
Atopy *	58 (55.2%)
**Comorbidities**	
Obesity (BMI ≥ 30 kg/m^2^)	24 (22.7%)
Chronic rhinosinusitis both allergic and not allergic	55 (57.75%)
Nasal polyposis	35 (33.3%)
Gastroesophageal reflux disease	29 (27.6%)
**Biomarkers**	
Peripheral Blood Eosinophils/μL	500 (340; 727.8)
Total IgE, IU/mL (*n* = 78)	182 (73.5; 406.5)
FeNO, ppb (*n* = 73)	41 (25, 61)
**Lung function**	
FEV_1_% predicted	63.7 ± 17.9
FVC% predicted	79.9 ± 16.2
FEV_1_/FVC%	67.5 ± 13
FEF25-75% predicted	32.7 ± 18.2
ACT (*n* = 98)	14.5 (11; 17.7)
Exacerbation history (previous year) (*n* = 84)	4.2 ± 2.5
**Maintenance therapy**	
ICS/LABA	105 (100%)
LAMA	63 (60%)
OCS	54 (51.43%)
LTRA	4 (3.80%)

Data are presented as: mean ± standard deviation (SD); median (IQR); *n* = (%). FeNO: fraction of exhaled nitric oxide; FEV1: forced expiratory volume in 1 s; FVC: forced vital capacity; FEF25-75: forced expiratory flow at 25–75% of forced vital capacity; ACT: asthma control test; ICS/LABA: inhaled corticosteroid/long-acting beta-agonist; OCS: oral corticosteroids; LAMA: long-acting muscarinic antagonists; LTRA: leukotriene receptor antagonists.* Atopy defined on the basis of skin prick test positivity.

## Data Availability

Data are available on request due to privacy restrictions.

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
