# Peer review of "Real-Life Effectiveness of Mepolizumab on Forced Expiratory Flow between 25% and 75% of Forced Vital Capacity in Patients with Severe Eosinophilic Asthma"

_biomedicines, 2021, doi:10.3390/biomedicines9111550_

Round 1

Reviewer 1 Report

The article Real-life effectiveness of mepolizumab on forced expiratory flow between 25% and 75% of forced vital capacity in patients with severe eosinophilic asthma” by Angelantonio Maglio et al. describes the findings about the influence of Mepolizumab on small airways of patients suffering from severe eosinophilic asthma. It attempts to explain the beneficial effect of the compound on asthma outcomes at the level of small airways. The article is easy to understand, the obtained results are well discussed.

However, the article needs corrections. First, there are missing data from patients included into the study; not for all patients data like smoking habits, asthma duration and age at onset, IgE, FeNO levels were included and it seems they have been somehow missing (see Table 1, n is not always equal to 105). Could the authors comment on that issue? Were there any inclusion/exclusion criteria for this study?

Second, the text and figures need extensive check and corrections. The authors use sometimes dots and sometimes commas for decimal fractions both in the text and in the figures, all of them should be changed to dots. Within the results, in the patients’ descriptions, there are missing units like for age or BMI, and that makes the text more difficult to read. Additionally, there are slightly other data in the text and in Table 1, probably due to approximations. There is also missing significance description in the legends of Figs. 2,3,4,6, and 7. Do different numbers of stars mean different levels of significance? In most cases, there is only one number of stars within the figure, so I believe, this should be corrected.

Author Response

We thank the reviewer for the appreciation and the precious comments. These are our poiny-by-point responses.

Q: However, the article needs corrections. First, there are missing data from patients included into the study; not for all patients data like smoking habits, asthma duration and age at onset, IgE, FeNO levels were included and it seems they have been somehow missing (see Table 1, n is not always equal to 105). Could the authors comment on that issue? Were there any inclusion/exclusion criteria for this study?

A: We are aware of the issue, considered this aspect among the limitations of the study, and specified in the manuscript that data on clinical and biological characteristics were not available for all patients due to different clinical records between participating centers, due to the nature retrospective of the study (lines 456-458). We have specified the inclusion criteria in the "study design" section of the Materials and Methods paragraph (lines 103-108): patients with a diagnosis of severe asthma according to European Respiratory society (ERS) / American Thoracic Society (ATS) guidelines and assessed for eligibility to mepolizumab treatment according to Italian Drug Agency (AIFA) prescription criteria, who had completed at least six months of therapy. We have decided not to exclude any patients, to ensure the "real-life" aspect as much as possible.

Q: Second, the text and figures need extensive check and corrections. The authors use sometimes dots and sometimes commas for decimal fractions both in the text and in the figures, all of them should be changed to dots. Within the results, in the patients’ descriptions, there are missing units like for age or BMI, and that makes the text more difficult to read. Additionally, there are slightly other data in the text and in Table 1, probably due to approximations. There is also missing significance description in the legends of Figs. 2,3,4,6, and 7. Do different numbers of stars mean different levels of significance? In most cases, there is only one number of stars within the figure, so I believe, this should be corrected.

A: We thank the reviewer  for the accuracy and the recommendations, and edited the text and figures providing all the missing data.

Reviewer 2 Report

It would be of interest to know if there is any correlation between baseline FEV1 and FEF25-75 and improvement in FEV1 and FEF25-75% at 18 months of the study.  If not, the results suggest that FEV25-75 may be a more sensitive marker for response than FEV1. 

Please  6 months the earliest time point of treatment; these patients had severe asthma and in most practices, would have been seen sooner than 6 months after starting this treatment.

Is there any bronchoscopy data showing onset of biologic effects so late after treatment start (> 6 months).

Is there any data on similar patients who opted to not be treated with a biologic; if so, this could provide a comparison group.

Consider discussing the significance of the use of a real world setting.  These patients were not in a rigorous controlled trial and showed improvement in outcomes.  

There is some incorrect wording throughout that needs to be corrected.  For example, eliminate "a" prior to "control" in lines 38 and 39, change "in" to "as" in line 43, "referring" to "referred" and "of" to "in" in line 97, "any" to "no" in line 170, line 188 is awkward; line 211 needs to be reworded (the response was first measured at 6 months and that is not rapid change to treatment with mepolizumab had a positive impact", change "over" to "in" in lline 281, eliminate "nowadays" in line 417, change "anyway to however" in line 420.   

Change commas to periods in the r and p values of figures 1, and 5, * need to be defined in the figures, correct "No" (with o as superscript) in panel c of figure 2 (I assume this stands for Number and I would spell out number or use # or change superscript o to regular script).

Author Response

We thank the reviewer for his interest, considerations and comments. These are our poiny-by-point responses.

Q: It would be of interest to know if there is any correlation between baseline FEV1 and FEF25-75 and improvement in FEV1 and FEF25-75% at 18 months of the study.  If not, the results suggest that FEV25-75 may be a more sensitive marker for response than FEV1. 

A: This is an important insight into the drug's effects on lung function. We found no correlation between baseline FEV1 and FEF25-75 and improvement in FEV1 and FEF25-75 at 18 months of the study. We think that further studies are needed focusing on this aspect.

Q: Please 6 months the earliest time point of treatment; these patients had severe asthma and in most practices, would have been seen sooner than 6 months after starting this treatment.

A: Patients have been strictly followed after starting treatment with mepolizumab, however in our study we took in consideration data collected from 6 months of treatment since the complete evaluation of those patients was performed at 6 months in order to consider them as “responders” to mepolizumab.

Q: Is there any bronchoscopy data showing onset of biologic effects so late after treatment start (> 6 months).

A: No data about bronchoscopy were available for our study population, and to our knowledge there is a lack of similar data in literature.

Q: Is there any data on similar patients who opted to not be treated with a biologic; if so, this could provide a comparison group.

A: Actually we do not have data regarding not biologics-treated severe asthmatic patients referred to the centers participating in our study, since all our severe asthmatic patients have met the requirements to a biological therapy prescription. However, this suggestion could represent a starting point for a future study.

Q: Consider discussing the significance of the use of a real world setting.  These patients were not in a rigorous controlled trial and showed improvement in outcomes.  

A: We appreciate the suggestion, and we highlighted this aspect in lines 412-418.

Q: There is some incorrect wording throughout that needs to be corrected.  For example, eliminate "a" prior to "control" in lines 38 and 39, change "in" to "as" in line 43, "referring" to "referred" and "of" to "in" in line 97, "any" to "no" in line 170, line 188 is awkward; line 211 needs to be reworded (the response was first measured at 6 months and that is not rapid change to treatment with mepolizumab had a positive impact", change "over" to "in" in lline 281, eliminate "nowadays" in line 417, change "anyway to however" in line 420.
Change commas to periods in the r and p values of figures 1, and 5, * need to be defined in the figures, correct "No" (with o as superscript) in panel c of figure 2 (I assume this stands for Number and I would spell out number or use # or change superscript o to regular script).

A: We thank the reviewer for the accuracy and the recommendations, and edited the text and figures providing all the changes.

Reviewer 3 Report

With real interest, I read the manuscript biomedicines-1417657. Although as a real-life study, it has several limitations compared to controlled trials, those are counterbalanced and even outnumbered/overridden by the very interesting results.

I have several suggestions and a few minor comments only.

Suggestions:

1. In your article, please, refer to PMID: 34572466, a very interesting article just published in this very special issue of Biomedicines.

2. In addition, in your work, please, refer to PMID: 33926084, another very interesting article recently published in the field.

3. Finally, your study group is very well characterized, especially as for a real-life study. What interests me the most, is obesity. Have any subgroup analyses after BMI-based substratification been performed? Could obesity affect the effects of the treatment, especially in your type 2 high subjects? In any case, please, shortly mention the potential meaning of obesity in asthma (PMID: 30057383 and 34576307).

Minor comments:

1. Please, unify (subscript/no subscript, etc.) the way you write “FEF 25-75“ throughout the manuscrpt.

2. Please, use dots not commas for decimals throughout the manuscrpt (incl. figures, tables, supplements).

3. While showing SD, only upper error bar be shown (suggestion only).

4. To stay uniform, Please, correct other details, e.g. in Figure 3a: “median“ -> “Median“.

Author Response

We thank the reviewer for his interest, consideration and comments. Here are our point-by-point responses.

Q: In your article, please, refer to PMID: 34572466, a very interesting article just published in this very special issue of Biomedicines.

In addition, in your work, please, refer to PMID: 33926084, another very interesting article recently published in the field.

A: We agree with the valuable suggestions and in this regard we have read with interest and included in the study the recommended citations (ref. #9, ref. #23).

Q: Finally, your study group is very well characterized, especially as for a real-life study. What interests me the most, is obesity. Have any subgroup analyses after BMI-based substratification been performed? Could obesity affect the effects of the treatment, especially in your type 2 high subjects? In any case, please, shortly mention the potential meaning of obesity in asthma (PMID: 30057383 and 34576307).

A: Thank you for your comments. We evaluated, in the context of comorbidities, the presence of obesity and its influence in our cohort, without however finding significant correlations. We have discussed this aspect more extensively (lines 370-374), and we have added the suggested references.

Q: Minor comments
1. Please, unify (subscript/no subscript, etc.) the way you write “FEF 25-75“ throughout the manuscrpt.
2. Please, use dots not commas for decimals throughout the manuscrpt (incl. figures, tables, supplements).
3. While showing SD, only upper error bar be shown (suggestion only).
4. To stay uniform, Please, correct other details, e.g. in Figure 3a: “median“ -> “Median“.

A: We appreciated the attention and precision in reporting the inconsistencies, and we corrected them based on the suggestions received.